# Effect of Flywheel versus Traditional Resistance Training on Change of Direction Performance in Male Athletes: A Systematic Review with Meta-Analysis

**DOI:** 10.3390/ijerph19127061

**Published:** 2022-06-09

**Authors:** Helmi Chaabene, Adrian Markov, Olaf Prieske, Jason Moran, Martin Behrens, Yassine Negra, Rodrigo Ramirez-Campillo, Ulrike Koch, Bessem Mkaouer

**Affiliations:** 1Department of Sports and Health Sciences, Faculty of Human Sciences, University of Potsdam, 14469 Potsdam, Germany; 2High Institute of Sports and Physical Education of Kef, University of Jendouba, Jendouba 8189, Tunisia; 3Division of Training and Movement Sciences, Research Focus Cognition Sciences, University of Potsdam, 14469 Potsdam, Germany; adrian.markov@uni-potsdam.de (A.M.); ulli-koch@gmx.de (U.K.); 4Division of Exercise and Movement, University of Applied Sciences for Sports and Management Potsdam, 14471 Potsdam, Germany; prieske@fhsmp.de; 5School of Sport, Rehabilitation and Exercise Sciences, University of Essex, Colchester CO4 3SQ, UK; jmorana@essex.ac.uk; 6Department of Sport Science, Institute III, Otto-von-Guericke University Magdeburg, 39104 Magdeburg, Germany; martin.behrens@ovgu.de; 7Department of Orthopedics, University Medicine Rostock, Doberaner Straße 142, 18055 Rostock, Germany; 8Research Unit (UR17JS01) Sport Performance, Health & Society, Higher Institute of Sport and Physical Education of Ksar Saïd, University of Manouba, Manouba 2010, Tunisia; yassinenegra@hotmail.fr; 9Exercise and Rehabilitation Sciences Laboratory, School of Physical Therapy, Faculty of Rehabilitation Sciences, Universidad Andres Bello, Santiago 7591538, Chile; rodrigo.ramirez@unab.cl; 10Department of Individual Sports, Higher Institute of Sport and Physical Education of Ksar Said, University of Manouba, Manouba 2010, Tunisia; bessem_gym@yahoo.fr

**Keywords:** human physical conditioning, eccentric training, strength training, athletes, sports, muscle strength

## Abstract

Objective: This study aimed to systematically review and meta-analyze the effect of flywheel resistance training (FRT) versus traditional resistance training (TRT) on change of direction (CoD) performance in male athletes. Methods: Five databases were screened up to December 2021. Results: Seven studies were included. The results indicated a significantly larger effect of FRT compared with TRT (standardized mean difference [SMD] = 0.64). A within-group comparison indicated a significant large effect of FRT on CoD performance (SMD = 1.63). For TRT, a significant moderate effect was observed (SMD = 0.62). FRT of ≤2 sessions/week resulted in a significant large effect (SMD = 1.33), whereas no significant effect was noted for >2 sessions/week. Additionally, a significant large effect of ≤12 FRT sessions (SMD = 1.83) was observed, with no effect of >12 sessions. Regarding TRT, no significant effects of any of the training factors were detected (*p* > 0.05). Conclusions: FRT appears to be more effective than TRT in improving CoD performance in male athletes. Independently computed single training factor analyses for FRT indicated that ≤2 sessions/week resulted in a larger effect on CoD performance than >2 sessions/week. Additionally, a total of ≤12 FRT sessions induced a larger effect than >12 training sessions. Practitioners in sports, in which accelerative and decelerative actions occur in quick succession to change direction, should regularly implement FRT.

## 1. Introduction

Change of direction (CoD) speed is a key determinant for successful performance in many team (e.g., soccer, rugby, and handball) [1,2,3,4] and individual (e.g., tennis and taekwondo) [5,6] sports. The ability to quickly decelerate and re-accelerate in a new direction is an important aspect that underpins a fast execution of a CoD task [7,8]. In soccer, for example, players can perform up to 600 cutting movements between 0° and 90° during a match [9]. Additionally, there is evidence that CoD tests can differentiate elite from sub-elite soccer players [10,11,12], representing a practically useful tool for performance monitoring and talent scouts [10,12]. As such, for the sake of improving sports performance and, ultimately, success in competition, it is important to develop CoD performance.

Performing CoD tasks engages multiple factors that are technical, biomechanical, anthropometric, and neuromuscular in nature [7,8]. Leg muscle quality, which includes reactive strength, concentric strength, and asymmetries, seems to be a crucial determinant of CoD performance [7]. Chaabene, Prieske, Negra, and Granacher [8] recently added eccentric muscle strength as another important determinant contributing to CoD performance. Overall, to optimize CoD performance, the ability to rapidly accelerate and decelerate the human body through concentric as well as eccentric muscle actions is crucial [8]. In this regard, traditional resistance training (i.e., strength training with combined concentric and eccentric muscle actions (TRT)) is an effective means to improve leg muscle quality in different populations, including youth and young adult athletes [13,14,15]. Over the past two decades, the effects of TRT on CoD performance in youth and young adult athletes have been extensively studied [14,16]. Lesinski, Prieske, and Granacher [13] conducted a meta-analysis on the effects of TRT on CoD performance in youth athletes aged between 6 and 18 years. The results indicated moderate effects of TRT (standardized mean difference (SMD) = 0.68) on CoD performance. However, that study did not include any recommendations that inform the training prescription to enhance CoD performance [13]. The results of another meta-analysis on the effects of TRT on CoD performance in youth and young physically active and athletic adults indicated a large effect (SMD = 0.82) [16]. In the same study, findings revealed that higher compared with lower resistance training volumes, frequencies, and intensities seemed not to have an additional effect on the magnitude of CoD performance improvements.

Alongside TRT, flywheel resistance training (FRT) is effective at improving CoD performance [8,17]. In a recent meta-analysis, Liu, Liu, Clarke, and An [17] indicated that interventions using flywheel inertial devices, among others, are effective to improve CoD performance (SMD = 1.35) in young and adult trained individuals. However, the heterogeneity across the eleven included studies was very high (I^2^ = 99.7%), thus making it difficult to derive clear recommendations. Likewise, Raya-González, et al. [18] compared the effects of FRT with TRT on CoD performance in athletic and healthy active individuals. The reported results indicated larger effects of FRT on CoD performance. However, these findings were rather preliminary, as the authors included just three studies, underlining the small body of evidence that has made it impossible to draw definitive conclusions. Additionally, for the comparator, the authors of the two previous meta-analysis [17,18] combined studies that included active control and a second intervention (e.g., TRT). All these limitations undermine the consistency of the findings.

In this regard, the question of the best training modality to improve CoD performance still needs to be clarified. More specifically, the effects of FRT vs. TRT on CoD performance have not yet been comprehensively and systematically assessed, highlighting a gap in the literature. Therefore, this study aimed to systematically review and meta-analyze the literature that contrasts the effects of FRT and TRT on CoD performance in male athletes. The second aim was to identify the main training-related variables that are associated with better CoD performance adaptations to help guid training prescription.

## 2. Methods

This systematic review was conducted according to the preferred reporting items for systematic review and meta-analysis (PRISMA) statements [19].

### 2.1. Literature Search

The electronic databases PubMed, Web of Science, Cochrane Library, Google Scholar, and SPORTDiscus were searched with no date restriction up to December 2021. Only peer-reviewed studies written in English were included. Keywords were collected through experts’ opinions, literature review, and controlled vocabulary (e.g., Medical Subject Headings (MeSH)). The following Boolean search syntax was used: (“Resistance training” OR “eccentric training” OR “flywheel training” OR “flywheel inertial resistance training” OR “flywheel isoinertial training” OR “flywheel overload training” OR “flywheel resistance training” OR “inertial training” OR “eccentric overload” OR “accentuated eccentric” “eccentric muscle action” OR “lengthening contraction” OR “eccentric exercise” OR “eccentric contraction” OR “negative work”) AND (“change of direction” OR “agility”) NOT (“elderly” OR “older adults” OR “patient” OR “disease”). Search results were screened by two researchers (HC and UK). Additionally, the reference lists of earlier published review articles on the topic were screened to search for further potentially relevant studies. An overview of the systematic search process is displayed in Figure 1.

### 2.2. Selection Criteria

To rate studies for eligibility, a PICOS (participants, intervention, comparators, study outcomes, and study design) approach was used [19]. The respective inclusion/exclusion criteria are displayed in Table 1.

### 2.3. Study Coding and Data Extraction

Two independent reviewers (UK and HC) extracted data from the included studies in a standardized template created with Microsoft Excel. In the case of disagreement regarding data extraction and study eligibility, co-author BM was consulted for clarification. To extract data (i.e., means and standard deviations) from figures, the WebPlotDigitizer software (https://apps.automeris.io/wpd/ (accessed on 15 April 2021)) was used [21]. The characteristics of the included studies are displayed in Table 2.

### 2.4. Study Quality

The Physiotherapy Evidence Database (PEDro) scale was used to evaluate the methodological quality of the included studies. Two authors (HC and YN) independently scored the included articles. The validity and reliability of the PEDro scale have been established previously [29,30]. Additionally, its agreement with other scales (e.g., Cochrane risk of bias tool) has been reported [31]. The internal validity of the included studies was rated on a scale from 0 (high risk of bias) to 10 (low risk of bias). A score of ≥6 represents the threshold for studies with a low risk of bias [30] (Table 3). Additionally, to estimate publication bias, a funnel plot was used. Further, to assess the presence of funnel plot asymmetry quantitatively, Egger’s regression test was used [32].

### 2.5. Statistical Analyses

To examine the effects of FRT vs. TRT on CoD performance, weighted between-group standardized mean differences (SMDs) were computed for pre-test and post-test values of each study using the following equation: SMD = (M_1_ − M_2_)/S_pooled_, where M_1_ is the mean pre/post-value of the FRT group, M_2_ is the mean pre/post-value of the TRT group, and S_pooled_ is the pooled standard deviation. To control for sample size, SMDs were adjusted according to the following equation, $\left(1-\frac{3}{4N-9}\right)$, with N representing the total sample size [33]. Additionally, baseline-adjusted SMD values were calculated as the difference between the pre-test SMD to post-test SMD [34]. The within-group effect sizes were calculated using the mean pre- and post-value of each of the FRT and TRT groups. Single training factor analyses were computed for training duration (6 weeks/8 weeks), training frequency (≤2/>2/sessions/week), and the total number of training sessions (≤12/>12 sessions). A random-effects model was used to weight each study and to determine the SMDs that are presented alongside 95% confidence intervals. The SMDs were interpreted using the conventions outlined by Cohen [35] (<0.2 “trivial”; ≤0.2 SMD < 0.5 “small”, ≥0.5 SMD < 0.8 “moderate”, ≥0.8 “large”). In addition, independent subgroup analyses were calculated for the single training variables (i.e., training duration, training frequency, and total number of training sessions). For all calculations, positive values were used to express performance gains. The level of between-study heterogeneity was assessed using the *I*^2^ statistic. This indicates the proportion of effects that are caused by heterogeneity as opposed to chance [19]. Low, moderate, and high heterogeneity correspond to *I*^2^ outcomes of 25, 50, and 75%, respectively [36]. A value above 75% is rated as being considerably heterogeneous [37]. Statistical calculations were conducted using R (version 4.1.0). The level of significance was set at *p* ≤ 0.05.

## 3. Results

### 3.1. Study Characteristics

Our literature search resulted in 1107 studies from which 35 potentially eligible articles were identified after removing duplicates and excluding studies based on titles and abstracts (Figure 1). A closer check identified 4 studies with missing data, 16 studies with no TRT group, 5 studies that did not assess CoD speed, 2 studies that did not include FRT, and 1 study that conducted FRT for upper limbs. Finally, seven studies were eligible for inclusion with a total of 16 experimental groups. The number of participants across the experimental groups ranged from 6 to 20 with a total of 201 (Table 2). The age of participants ranged from 13 to 24 years. All participants across the included studies were recruited from specific sports settings (e.g., sports clubs or teams) and were actively participating in competitive events. Therefore, they can be categorized as athletes [20]. The training duration across the included studies lasted between 6 and 8 weeks. The training frequency ranged between one and three sessions per week. The total number of training sessions ranged between 8 and 17.

The median PEDro score of the included studies was 6 (range 3 to 6). Six out of the seven included studies reached the cut-off value ≥ 6 (Table 3). The visual inspection of the funnel plot indicated a symmetrical distribution pattern of the effects, illustrating the absence of publication bias (Figure 2). This is strengthened by Egger’s regression outcome, which indicated that the distribution pattern of the effect in the funnel plot is symmetrical (*t* = 0.36, *p* = 0.727).

### 3.2. Between-Group Effects

The effects of FRT vs. TRT are displayed in Figure 3. There is a significant difference between the effects of FRT and TRT in favor of the former (SMD = 0.64 [0.06 to 1.21]; *p* = 0.034). The between-study heterogeneity was moderate and significant (I^2^ = 65% [28.7% to 82.8%]; *p* = 0.003).

### 3.3. Within-Group Effects

The within-group effects indicated significant large effects of FRT on CoD performance (SMD = 1.63 [0.35 to 2.91]; *p* = 0.019; Figure 4). The between-study heterogeneity was high and significant (I^2^ = 86.5% [76.4% to 92.3%]; *p* < 0.01). For TRT, significant moderate effects were observed (SMD = 0.62 [−0.004 to 1.248]; *p* = 0.051; Figure 5). The between-study heterogeneity was moderate and significant (I^2^ = 65.9% [30.8% to 83.2%]; *p* = 0.002).

### 3.4. Single Training Factor Analyses

For FRT, no significant effects for 8 weeks (SMD = 1.15 [−0.50 to 2.82]; *p* > 0.05) and 6 weeks (SMD = 2.05 [−0.61 to 4.71]; *p* > 0.05) of training were observed. In terms of training frequency, ≤2 sessions/week resulted in significant large effects (SMD = 1.33 [0.32 to 2.35]; *p* < 0.05), whereas no significant effects of >2 sessions/week were noted (SMD = 2.35 [−5.04 to 9.74]; *p* > 0.05). Regarding the total number of training sessions, the results indicated significant large effects of ≤12 (SMD = 1.83 [0.60 to 3.06]; *p* < 0.05), with no significant effects of >12 (SMD = 1.52 [−1.36 to 4.39]; *p* > 0.05) sessions. No significant differences between all single training factors were detected (*p* > 0.05) (Table 4).

With respect to TRT, the findings indicated no significant effects of 8 weeks (SMD = 0.55 [−0.34; 1.45]; *p* > 0.05) and 6 weeks of training (SMD = 0.65 [−0.67; 1.98]; *p* > 0.05). A training frequency of either ≤2 sessions/week or >2 sessions/week generated no significant effects (SMD = 0.43 [−0.01; 0.87] and SMD = 0.95 [−2.60; 4.51] both *p* > 0.05, respectively). With respect to the total number of training sessions, no significant effects of ≤12 (SMD = 0.55 [−0.41; 1.52]; *p* > 0.05) as well as >12 (SMD = 0.67 [−0.62; 1.97]; *p* > 0.05) sessions were observed. The differences between all single training factors were not significant (*p* > 0.05) (Table 4).

## 4. Discussion

The aims of this systematic review with meta-analysis were (i) to examine the effects of FRT vs. TRT on CoD performance in male athletes and (ii) to identify the main training variables that are associated with better CoD performance adaptations in response to these types of training. The main findings indicated an advantage of FRT over TRT on CoD performance in male athletes. This has been reinforced by the within-group analysis, which showed larger effects of FRT compared with TRT on CoD performance. Independently computed single training factor analyses for FRT indicated that ≤2 sessions/week resulted in larger effects than >2 sessions/week. Additionally, the results indicated that ≤12 FRT sessions induced larger effects than >12 training sessions.

### 4.1. Primary Analysis

Our findings showed a significant effect difference between FRT and TRT in favor of FRT (SMD = 0.64). This highlights an advantage of FRT over TRT on CoD performance in male athletes. The previous result was substantiated by the within-group analysis, where FRT induced large improvements (SMD = 1.63), while TRT resulted in moderate enhancements (SMD = 0.64) in CoD performance. There is evidence indicating that FRT can provide an eccentric overload stimulus [38,39,40,41]. It is, however, worth noting that several factors moderate the level of eccentric overload achieved by the flywheel device, such as the inertial load used [39,40], the adopted technique [42], and the preceding concentric output (i.e., concentric velocity) [38,43]. CoD performance is determined by multiple factors, amongst which the eccentric strength of the thigh muscles plays a key role [7,8,44,45,46]. Specifically, the eccentric muscle strength influences the braking phase (i.e., deceleration) during a rapid CoD task and, therefore, facilitates an earlier re-acceleration in a different direction [8,47]. Over the past decade, a body of evidence has emerged highlighting the benefits of FRT from a morphological (e.g., improved muscle mass/size) [41,48], neuromuscular (increased electromyography activity) [49], and physical fitness perspectives [8,25,41,45,50,51,52,53].

In this regard, persuasive evidence from cross-sectional works indicated moderate-to-large associations between eccentric muscle strength and CoD performance [47,54,55,56,57]. Jones, et al. [58] examined the impact of the eccentric muscle strength of the knee extensors in female soccer players aged 22 years. The authors reported that greater eccentric strength is associated with faster CoD performance. Additionally, the same authors revealed that players with higher eccentric strength displayed better deceleration capabilities during the penultimate ground contact during faster movement velocities. Alongside cross-sectional studies, findings from intervention studies indicated that CoD tasks seem to largely benefit from eccentric overload exercises, possibly due to a performer’s ability to store elastic energy that can be efficiently reutilized in subsequent accelerative movements [59]. Indeed, all the studies included in this systematic review showed positive effects of FRT on CoD performance, highlighting the robustness of the effect across the range of included studies. For example, Coratella, et al. [23] evaluated eight weeks of FRT (inertia = 0.11 kg⋅m^−2^) vs. TRT using free weights (80% one-repetition maximum (1RM)) during the squat exercise on CoD performance (i.e., T-test) in young male soccer players aged 23 years. These researchers observed large improvements in CoD performance (effect size (ES) = 1.44) following FRT with no significant effects of TRT (ES = 0.33). Recently, Stojanović, et al. [26] studied the effects of FRT using an isoinertial flywheel device (inertia = 0.075 kg⋅m^−2^) vs. TRT using free weights (80% 1RM) on CoD performance (T-test) in male basketball players aged 18 years. These authors revealed significantly larger improvements following FRT (ES = 2.78) compared with TRT (ES = 1.64). Overall, greater eccentric strength seems to facilitate faster CoD performance by improving the ability to tolerate the greater loads associated with faster approach velocities, more particularly, during the penultimate and final foot contacts during movement [58,60,61]. Moreover, higher eccentric strength increases joint stability and facilitates better force transfer through joints, all of which contribute to more efficient CoD abilities [62].

In the same context, it has been demonstrated that FRT has the potential to improve muscle power [25,50]. Of note, earlier findings in youth male team players indicated large-to-very large associations between muscle power and CoD performance [63]. All the included studies, except for one [28], that involved measures of muscle power (e.g., countermovement jump) alongside CoD tasks demonstrated significant enhancements following FRT. Particularly, the majority of the studies [24,25,26,27,64] indicated larger muscle power improvements following FRT compared with TRT. This is in line with recent findings from a systematic review and meta-analysis in which the authors reported larger improvements in jumping performance following FRT compared with TRT in healthy physically active and athletic individuals [18]. Indeed, increasing muscle power can contribute to a greater braking impulse and reduces braking and contact times, facilitating a rapid transition into the propulsion (i.e., re-acceleration) phase of a given rapid movement [58,61]. In sum, FRT seems to be more effective than TRT in improving CoD performance in male athletes.

### 4.2. Single Training Factor Analysis

Independent single training factor analyses were undertaken for both FRT and TRT. For FRT, the results showed larger effects for ≤2 sessions/week compared with >2 sessions/week. Similar findings were reported by Chaabene, Prieske, Moran, Negra, Attia, and Granacher [16]. More specifically, the authors revealed that higher compared with lower frequencies (i.e., 2 vs. 3 sessions/week) of TRT have no additive effects on CoD performance in physically active and athletic adults. Tesch, et al. [65] synthesized the results of a number of FRT studies and indicated that no more than two sessions per week should be performed with 48 h of recovery between sessions. In addition, our results indicated larger effects of ≤12 FRT sessions compared with >12 training sessions. Based on our findings, it appears more beneficial to favor lower frequencies and a smaller number of total training sessions during FRT. Overall, these results have implications on training design for FRT in male athletes. It is worth noting though that our findings are rather preliminary and should be interpreted with caution given that only two out of the seven included studies used >2 sessions/week. This means that future investigations are needed to substantiate the current results. In terms of TRT, no significant effects of any of the training factors were reported.

### 4.3. Future Research Perspectives

CoD tasks provide a foundation for more advanced agility skills [66,67]. While the current findings help to inform the training prescription to improve CoD performance, future studies should address the effects of FRT on agility (i.e., rapid directional changes to an external stimulus) [7], which is more relevant for performance in competition [67]. Additionally, we have not found any study that was carried out in female participants, with all current studies only including male participants. This would undermine the applicability of the current findings to females. Therefore, future studies should also be carried out in female populations. Moreover, the duration of training across the included studies ranges between 6 and 8 weeks. We could not find any study that examined the effect of longer durations (i.e., >8 weeks) of FRT vs. TRT on CoD speed performance. As such, future studies with longer training durations are warranted.

### 4.4. Limitations

The first limitation of the present study is related to the limited number of included studies. This indicates that this research topic remains under-investigated, though we do see the current study as an adequate starting point in generating a consensus on the effects of FRT on CoD performance and a call to action for research in this particular area. In addition, the significant heterogeneity across the eligible studies could undermine the accuracy of the findings, though such heterogeneity is highly common in meta-analyses, meaning caution must be exercised in interpreting the results. Moreover, single training factor analyses were conducted independently and not interdependently. Such an analysis must be considered with caution, given that the training variables were considered as single factors regardless of the interdependency between them. On this, we dichotomized subgroup continuous data, and this could result in residual confounding and reduced statistical power when analyzing the reported results.

## 5. Conclusions

FRT appears to be more effective than TRT in improving CoD performance in male athletes. In addition, independently computed single training factor analyses for FRT indicated that ≤2 sessions/week resulted in larger effects than ≥2 sessions/week. Additionally, the results showed that a total of ≤12 FRT sessions induced larger effects than >12 training sessions. As such, it seems more beneficial to favor lower frequencies and a smaller number of total training sessions during FRT. The results of the present study can help coaches as well as strength and conditioning professionals to design better training interventions to improve CoD performance in male athletes.

## Figures and Tables

**Figure 1 ijerph-19-07061-f001:**
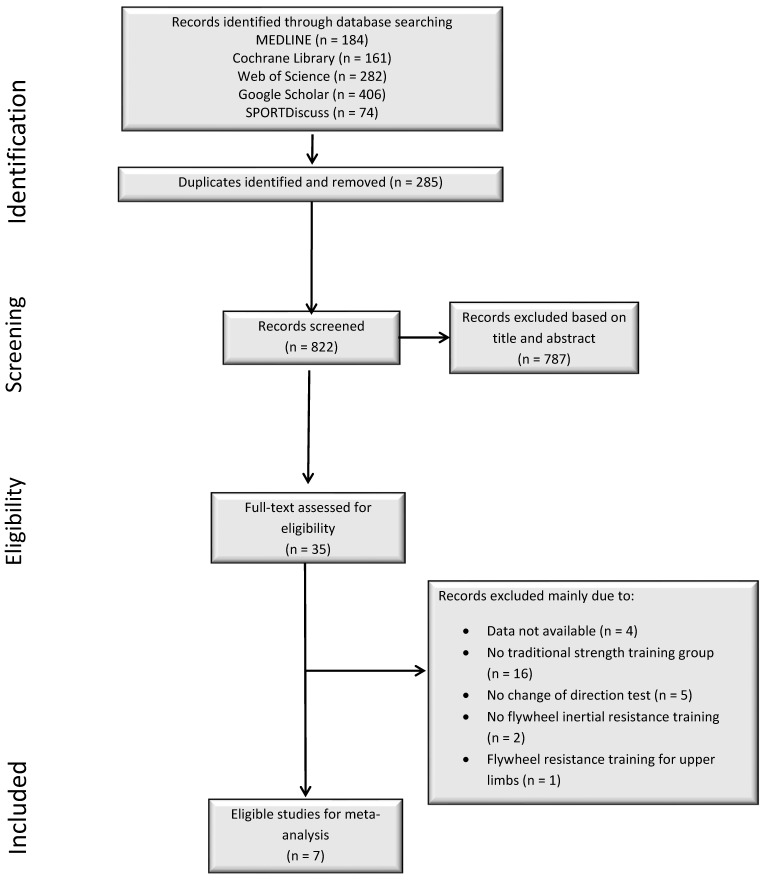
Flow chart illustrating the selection process for all included and excluded studies.

**Figure 2 ijerph-19-07061-f002:**
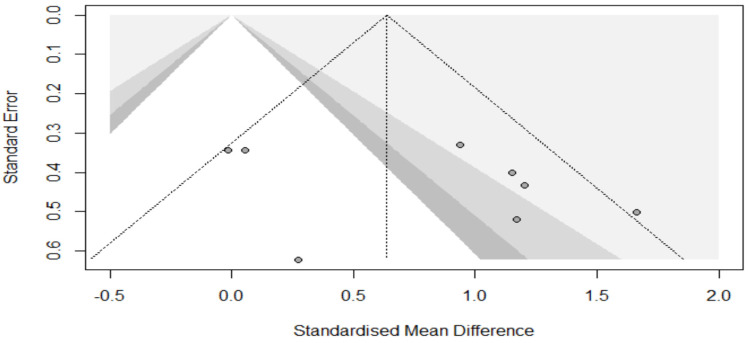
Funnel plot illustrating the symmetrical distribution of the effects across the included studies.

**Figure 3 ijerph-19-07061-f003:**
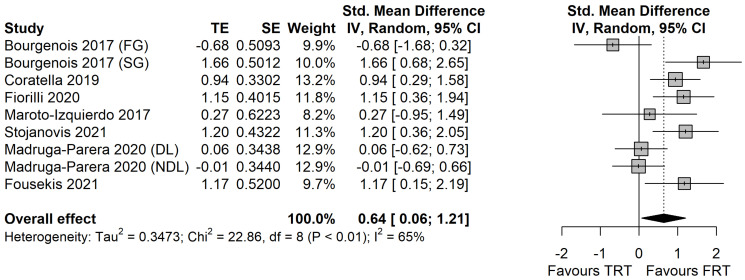
The effects of flywheel resistance training vs. traditional resistance training on change of direction performance in male athletes [22,23,24,25,26,27,28]. *FG: fast group; SG: slow group; DL: dominant leg; NDL: non-dominant leg; RL: right leg. TST: traditional strength training; AET: accentuated eccentric training.*

**Figure 4 ijerph-19-07061-f004:**
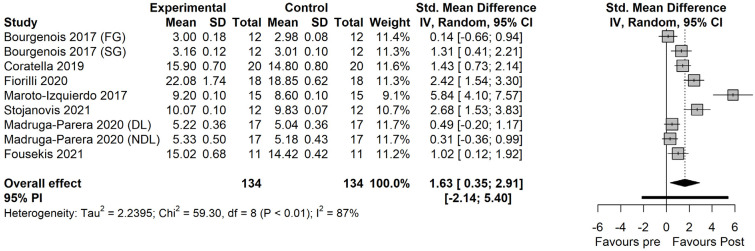
The effects of flywheel resistance training on change of direction performance in male athletes [22,23,24,25,26,27,28]. *FG: fast group; SG: slow group; DL: dominant leg; NDL: non-dominant leg; RL: right leg.*

**Figure 5 ijerph-19-07061-f005:**
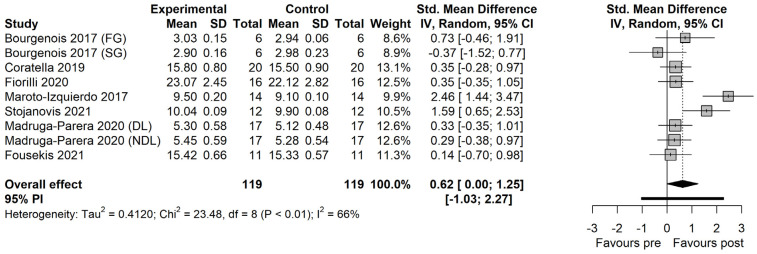
The effects of traditional resistance training on change of direction performance in male athletes [22,23,24,25,26,27,28]. *FG: fast group; SG: slow group; DL: dominant leg; NDL: non-dominant leg; RL: right leg.*

**Table 1 ijerph-19-07061-t001:** Selection criteria.

Category	Inclusion Criteria	Exclusion Criteria
**Population**	Youth and young male athletes *	Studies recruiting individuals with adverse health status (e.g., diabetes)
**Intervention**	Flywheel inertial resistance training (e.g., isoinertial exercises using flywheel)	Absence of resistance training using flywheel
**Comparator**	Traditional strength training program (i.e., strength exercises soliciting concentric/eccentric muscle actions)	Absence of a traditional strength training group
**Outcome**	Measures of CoD performance (e.g., T-test time, Illinois test time)	Measures of linear speed, lack of baseline and/or follow-up data
**Study design**	Randomized controlled trials or randomized cross-over trials	Quasi-experimental study design

* Training experience was determined with regard to the context from which participants were recruited. Athletes were recruited from specific sports settings (e.g., sports clubs or teams) and were actively participating in competitive events [20].

**Table 2 ijerph-19-07061-t002:** Characteristics of the included studies.

Study (Design)	Group	N	Age (Years)	Training Expertise	Body Mass (kg)	Body Height (m)	Description	Training Duration (Weeks)	Frequency (Session/Week)	Session Duration (min)	Volume	Intensity	Total Number of Training Sessions	CoD Protocol
Bourgeois, et al. [22] (Randomized cross-over trial)	FRT	12	15.0 ± 0.9	High school athletes	80.2 ± 15.3	1.8 ± 0.1	Upper and lower body isoinertial resistance exercises (3 s eccentric duration followed by concentric action “as fast as possible”)	6	3/wk	60	3 sets, 6–10 reps	NR	16	CoD 180° and 45°; modified 505 CoD test
	TRT	6 (former FRT)	15.3 ± 0.5	High school athletes	81.8 ± 12.4	1.8 ± 0.1	Upper and lower body isoinertial resistance exercises (no constraints on tempo)	6	3/wk	60	3 sets, 6–10 reps	NR	17	CoD 180° and 45°; modified 505 CoD test
Coratella, Beato, Cè, Scurati, Milanese, Schena, and Esposito [23] (RCT)	FRT	20	23 ± 4	Athletes	77 ± 5	1.80 ± 0.11	Squat using flywheel	8	1/wk	20	4–6 sets, 8 reps	flywheel squats inertia: 0.11 kg·m^−2^	8	T-test; 20 + 20 m shuttle test
	TRT (Weight training)	20	23 ± 4	Athletes	77 ± 5	1.80 ± 0.11	Squat using barbells	8	1/wk	20	6 sets, 8 reps	480% 1RM,	8	T-test; 20 + 20 m shuttle test
Fiorilli, Mariano, Iuliano, Giombini, Ciccarelli, Buonsenso, Calcagno, and di Cagno [24] (RCT)	FRT	18	13.21 ± 1.21	Athletes	51.25 ± 6.71	1.65 ± 0.10	Lower body isoinertial resistance exercises	6	2/wk	NR	2 ex, 4 sets, 7 reps	17 Borg’s Scale	12	Y-agility (45°), Illinois CoD test
	TRT/Plyo	16	13.36 ± 0.80	Athletes	52.10 ± 5.23	1.68 ± 0.07	Plyometric exercises	6	2/wk	NR	2 ex, 3–4 sets, 7–10 reps	17 Borg’s Scale	12	Y-agility (45°), Illinois CoD test
Maroto- Izquierdo, García-López, and de Paz [25] (RCT)	FRT	15	19.8 ± 1	Athletes	82.3 ± 3.3	1.86 ± 0.08	Flywheel resistance training with eccentric overload (leg press)	6	2–3/wk	NR	4 sets, 7 reps	Maximum-concentric effort	15	T-test
	TRT	14	23.8 ± 1.6	Athletes	85.6 ± 3.7	1.84 ± 0.01	Weight-stack machine (leg press)	6	2–3/wk	NR	4 sets, 7 reps	7RM	15	T-test
Stojanović, Mikić, Drid, Calleja- González, Maksimović, Belegišanin, and Sekulović [26] (RCT)	FRT	12	17.58 ± 0.52	Athletes	75.53 ± 5.43	190.54 ± 4.98	One-arm dumbbell row, rotational Pall of press, biceps curls + upright row complex, half squat on isoinertial device, Romanian deadlift on isoinertial device	8	1–2/wk	NR	2–4 sets, 8–15 reps	85% 1RM (except Rotational Pallof press)	12	T-test (Semenick)
	TRT	12	17.52 ± 0.58	Athletes	78.78 ± 8.01	190.58 ± 6.56	One-arm dumbbell row, rotational Pallof press, biceps curls + upright row complex, half squat with free weights, Romanian feadlift with free weights	8	1–2/wk	NR	2–4 sets, 8–15 reps	85% 1RM (except Rotational Pallof press)	12	T-test (Semenick)
Madruga-Parera, Bishop, Fort-Vanmeerhaeghe, Beato, Gonzalo-Skok, and Romero-Rodríguez [27] (RCT)	FRT	17	15.9 ± 1.4	Athletes	70.5± 13.3	1.74 ± 0.73	Isoinertial exercises (CoD drills, handball sport-specific exercises)	8	2	NR	3 sets, 8–12 reps	RPE (6–9)	16	CoD 180° Test
	TRT	17		Athletes			Cable resistance exercises (CoD drills, handball sport-specific exercises)	8	2	NR			16	
Fousekis, Fousekis, Fousekis, Manou, Michailidis, Zelenitsas, and Metaxas [28] (RCT)	FRT	11	24.0 ± 6.6	Athletes	77.0± 4.4	1.80 ± 0.42	Isoinertial training during semi-squatting using flywheel	6	2	NR	3–4 sets, 10 reps	NR	12	Illinois CoD test
	TRT	11	19.7 ± 2.1	Athletes	75.3 ± 3.9	1.80 ± 0.50	Semi-squat using free weights	6	2	NR	3–4 sets, 8–10 reps	75–85% 1RM	12	

RCT: Randomized controlled trial; RPE: The rating of perceived exertion; FRT: Flywheel resistance training group; TRT: Traditional resistance training group; 1RM: one-repetition maximum; N: Number.

**Table 3 ijerph-19-07061-t003:** Methodological quality of the included studies based on the physiotherapy evidence database (PEDro) scale.

Study	Eligibility Criteria	Randomized Allocation	Blinded Allocation	Group Homogeneity	Blinded Subjects	Blinded Therapists	Blinded Assessor	Drop Out <15%	Intention-to-Treat Analysis	Between-Group Comparison	Point Estimates and Variability	PEDro Score
Bourgeois, Gamble, Gill, and McGuigan [22]	●	○	○	○	○	○	○	○	●	●	●	3
Coratella, Beato, Cè, Scurati, Milanese, Schena, and Esposito [23]	●	●	●	○	○	○	○	●	●	●	●	6
Fiorilli, Mariano, Iuliano, Giombini, Ciccarelli, Buonsenso, Calcagno, and di Cagno [24]	●	●	○	●	○	○	○	●	●	●	●	6
Maroto-Izquierdo, García-López and de Paz [25]	●	●	○	●	○	○	○	●	●	●	●	6
Stojanović, Mikić, Drid, Calleja- González, Maksimović, Belegišanin, and Sekulović [26]	●	●	○	●	○	○	○	●	●	●	●	6
Madruga-Parera, Bishop, Fort- Vanmeerhaeghe, Beato, Gonzalo-Skok, and Romero-Rodríguez [27]	●	●	●	○	○	○	●	○	●	●	●	6
Fousekis, Fousekis, Fousekis, Manou, Michailidis, Zelenitsas, and Metaxas [28]	●	●	○	●	○	○	●	○	●	●	●	6

● adds a point on the score, ○ adds no point on the score. The item “eligibility criteria” is not included in the final score.

**Table 4 ijerph-19-07061-t004:** Single training factor analyses for the flywheel and traditional resistance training.

Subgroup	Nb Studies (Nb Exp)	Estimated Effect Size Mean (95%, CI)	Within-Subgroup *p*	Between-Subgroup *p*	Within Group *I^2^*
**Flywheel resistance training**					
**Training duration**					
6 weeks	4(5)	2.05 [−0.61 to 4.71]	*p* > 0.05	*p* = 0.414	90%
8 weeks	3(4)	1.15 [−0.50 to 2.82]	*p* > 0.05		81%
**Training frequency**					
≤2 sessions/week	5(6)	1.33 [0.32 to 2.35]	*p* < 0.05	*p* = 0.564	80%
>2 sessions/week	2(3)	2.35 [−5.04 to 9.94]	*p* > 0.05		94%
**Total number of training sessions**					
≤12 sessions	4(4)	1.83 [0.60 to 3.06]	*p* < 0.05	*p* = 0.774	63%
>12 sessions	3(5)	1.52 [−1.36 to 4.39]	*p* > 0.05		90%
**Traditional resistance training**					
**Training duration**					
6 weeks	4(5)	0.65 [−0.67; 1.98]	*p* > 0.05	*p* = 0.855	77%
8 weeks	3(4)	0.55 [−0.34; 1.45]	*p* > 0.05		50%
**Training frequency**					
≤2 sessions/week	5(6)	0.43 [−0.01; 0.87]	*p* > 0.05	*p* = 0.554	25%
>2 sessions/week	2(3)	0.95 [−2.60; 4.51]	*p* > 0.05		85%
**Total number of training sessions**					
≤12 sessions	4(4)	0.55 [−0.41; 1.52]	*p* > 0.05	*p* = 0.821	52%
>12 sessions	3(5)	0.67 [−0.62; 1.97]	*p* > 0.05		76%

CI: Confidence interval; Nb studies = number of studies; Nb Exp: number of experimental groups.

## Data Availability

Authors would be happy to provide the data upon reasonable request.

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
