# Peer review of "Effect of Flywheel versus Traditional Resistance Training on Change of Direction Performance in Male Athletes: A Systematic Review with Meta-Analysis"

_ijerph, 2022, doi:10.3390/ijerph19127061_

Round 1

Reviewer 1 Report

The review article focuses on the effects of flywheel training over traditional resistance training (TRT) on COD performance. The entire article was well-written, and all the results were fascinating. However, some parts should be clarified and explained carefully.

Major Comments:

1.               I recommend adding a table to describe single training factor analyses.

2.               The authors compared the training effects between ≤2/>2/sessions/week. However, only two out of seven studies observed >2/sessions/week, and the one reached below the cut-off value. This result should be discussed more in detail.

3.               This review concluded that it is more beneficial to favor lower frequencies and fewer total training sessions during FRT. I can understand the conclusion from the results. However, this result also means the fact that athletes should not do FRT for a long period of time (more than 8 weeks). The authors should discuss that in the long-term training adaptations. For example, many studies revealed that eccentric training for more than eight weeks is beneficial for athletes in terms of muscle strength.

4.               And/or the authors should add a period (e.g., less than 12 sessions “during eight weeks”)

Minor Comments:

1.      There are extra “spaces” in many places in the text (e.g., L78, L188, 239, 298, 309, 333).

2.      L36. Factors, not tactors.

3.      L161 and 185. According to Table 2, all participants in all selected articles were trained for 6 or 8 weeks. However, they trained for 5-8 weeks (L185). Please clarify. And, “>6-8 weeks” does not make sense. Please use “>6 weeks.”

4.      L203. What do “first group” and “slow group” mean?

Author Response

Dear Reviewer,

Thank you for your insightful and constructive comments on our manuscript. We very much value your input. We have considered all your concerns and included changes wherever needed. We feel that the integrated edits have certainly strengthened our manuscript.

Best regards,

Helmi Chaabene

Comments and Suggestions for Authors

The review article focuses on the effects of flywheel training over traditional resistance training (TRT) on COD performance. The entire article was well-written, and all the results were fascinating. However, some parts should be clarified and explained carefully.

Authors’ reply: Thank you for your affirmative comment. We have considered all your suggestions and concerns in the revised version of the manuscript.

Major Comments:

I recommend adding a table to describe single training factor analyses.

Authors’ reply: A table detailing the outcomes of the single training factor analyses was added as suggested (please, refer to table 4 in the revised version). Thank you.

               The authors compared the training effects between ≤2/>2/sessions/week. However, only two out of seven studies observed >2/sessions/week, and the one reached below the cut-off value. This result should be discussed more in detail.

Authors’ reply: Thank you for your comment. That’s correct. Only two studies applied more than two sessions per week. The reason why we consider this specific outcome rather preliminary and should be interpreted with caution. We have included the following statements in the discussion to highlight this aspect:

It is worth noting though that our findings are rather preliminary and should be interpreted with caution given that only two out of the seven included studies have used >2 sessions/week.  This means that future investigations are needed to substantiate the current results.”

               This review concluded that it is more beneficial to favor lower frequencies and fewer total training sessions during FRT. I can understand the conclusion from the results. However, this result also means the fact that athletes should not do FRT for a long period of time (more than 8 weeks). The authors should discuss that in the long-term training adaptations. For example, many studies revealed that eccentric training for more than eight weeks is beneficial for athletes in terms of muscle strength. And/or the authors should add a period (e.g., less than 12 sessions “during eight weeks”)

Authors’ reply: Thank you for your pertinent remark. The training duration across the included studies ranges between 6 and 8 weeks and the findings favored lower training frequencies and fewer total training sessions during FRT. Of note, such outcomes cannot be considered conclusive due to the limited number of studies included. This is mentioned in the limitations section of the manuscript. In fact, we have elaborated conclusions within the boundaries of our findings. However, we are not claiming that longer training durations or a higher total number of training sessions would be deleterious to performance. The point is that we don’t know as we don’t have any study available, to the best of our knowledge, that looked at the effects of longer durations (i.e., >8 weeks) of FRT on CoD speed. We could think of fatigue as one of the factors that favor lower training frequencies and fewer training sessions during FRT. More specifically, FRT tends to provide an eccentric overload. Such an overload is associated with increased fatigue and delayed muscle soreness, which together could lead to a decrease in performance. In this context, Tesch, et al. (2017) recommended performing no more than two sessions per week of FRT with 48 h of recovery between sessions. Overall, this research topic remains under-investigated, the reason why we do consider the current study as an adequate starting point in generating a consensus on the effects of FRT on CoD performance, and a call to action for research in this particular area. The following statements were added to the future research perspective section of the paper:

“Moreover, the duration of training across the included studies ranges between 6 to 8 weeks. We could not find any study that examined the effect of longer durations (i.e., >8 weeks) of FRT vs. TRT on CoD speed performance. As such, future studies with longer training durations are warranted.”

Tesch, P.A.; Fernandez-Gonzalo, R.; Lundberg, T.R. Clinical Applications of Iso-Inertial, Eccentric-Overload (YoYo™) Resistance Exercise. Frontiers in physiology 2017, 8, 241, doi:10.3389/fphys.2017.00241.

Minor Comments:

There are extra “spaces” in many places in the text (e.g., L78, L188, 239, 298, 309, 333).

Authors’ reply: All extra spaces were removed. Thank you.

L36. Factors, not tactors.

Authors’ reply: This was a typo, now corrected. Thank you

L161 and 185. According to Table 2, all participants in all selected articles were trained for 6 or 8 weeks. However, they trained for 5-8 weeks (L185). Please clarify. And, “>6-8 weeks” does not make sense. Please use “>6 weeks.”

Authors’ reply: Thank you for raising this to our attention. The duration of the training interventions throughout the included studies is either 6 or 8 weeks. We have specified in the revised version that the training duration was either 6 or 8 weeks. Apologies for the mistake.

L203. What do “first group” and “slow group” mean?

Authors’ reply: In the original study of Bourgeois et al. (2017), participants were assigned to faster group and slower group based on the group median of total sprint times (calculated from the before-intervention data set), to examine category differences. Thank you.

Reviewer 2 Report

I attached the manuscript with some considerations.

Author Response

Dear Reviewer,

Thank you for your insightful and constructive comments on our manuscript. We very much value your input. We have considered all your concerns and included changes wherever needed. We feel that the integrated edits have strengthened our manuscript.

Best regards,

Helmi Chaabene

Comments

factors?

Authors’ reply: Correct. This was a typo. Thank you

Is there one space more?

Authors’ reply: All extra spaces were removed throughout the manuscript

Perhaps, you should rewrite the sentence because the verb "make" appears twice times.

 Authors’ reply: The sentence was revised as follows: “However, these findings are rather preliminary as the authors included just three studies, underlining the small body of evidence that has made it impossible to draw definitive conclusions.”

Change the connector.

 Authors’ reply: Changed to “in this regard”. Thanks.

“The results of such a study would help coaches and strength and conditioning professionals to design better training interventions to improve CoD performance.”

I think that this sentence should be written in other place, for example, in the conclusion where you have written about the practical application for this study.

Authors’ reply: This sentence was moved to the conclusion section. Thank you.

Methods

"change of direction" is repeated

Authors’ reply: The duplicate was removed. Thank you.

I think that this figure should be described in the results, because the criteria stablished is described before.

Authors’ reply: We have provided a more detailed description of the flow chart in the results section as follows: “Our literature search resulted in 1107 studies from which thirty-five potentially eligible articles were identified after removing duplicates and excluding studies based on titles and abstracts (Fig 1). A closer check identified four studies with missing data, 16 studies with no TRT group, five studies that did not assess CoD speed, two studies that did not include FRT, and one study that conducted FRT for upper limbs.”

fig 1, table 2 & 3, are showed before than their explanation. So, you have to change the order from these figures and tables.

Authors’ reply: We have indicated the correct location of tables and figures. Thank you.

should play?

Authors’ reply: It is actually correct. “Play” refers to competitive “competitive play”.

Each reference has a different type when journal is cited. Please, put all the journals with the same type.

Authors’ reply: Thank you for your comment. This was fixed and journal citation follows a standard style in the revised version.

Reviewer 3 Report

I would suggest proofreading and correcting the English language and style.

Author Response

I would suggest proofreading and correcting the English language and style.

Authors’ reply: Dear Reviewer, thank you for your positive feedback. We carefully double-checked the quality of writing and included a number of corrections and edits.

Reviewer 4 Report

General Comments

This is interesting information on a relatively new training apparatus.   

Minor Comments:

  Line                                                                      Comments

      27                      Should that be “…review and meta-analyze the effects ….”?

      28                      Should “over” be rephrased as “versus” so as not to bias to the review?

      34                      Would it help the reader to indicate “….large effect of ≤12 sessions…”?

      36                      “…any of the training factors…….”

      66                      Should that read “….young adult athletic individuals……”?

      94                      Should that again be “…review and meta-analyze the literature ….”?

References         Be sure to check the correct journal notation of journal titles.  Some are capitalized and some are not. 

Author Response

Dear Reviewer,

Thank you for your helpful and valuable comments on our manuscript. We very much value your input. We have considered all your concerns and included changes wherever needed. We feel that the integrated edits have strengthened our manuscript.

Best regards,

Helmi Chaabene

General Comments

This is interesting information on a relatively new training apparatus.   

 Authors’ reply: We highly appreciate your positive feedback. Thank you.

Minor Comments:

  Line Comments

      27 Should that be “…review and meta-analyze the effects ….”?

 Authors’ reply: Thank you for your comment. It depends on whether to use American (analyze) or British (analyse) English. So basically, both are correct. In the present work, we have decided to use British English.

      28 Should “over” be rephrased as “versus” so as not to bias to the review?

 Authors’ reply: Corrected as suggested. Thank you.

      34 Would it help the reader to indicate “….large effect of ≤12 sessions…”?

 Authors’ reply: For better readability and clarity, we have edited the statement as follows: Additionally, significant large effect of ≤12 FRT sessions (SMD=1.83) was observed with no effect of >12 sessions.”

      36 “…any of the training factors…….”

 Authors’ reply: The typo was fixed. Thank you

      66 Should that read “….young adult athletic individuals……”?

 Authors’ reply: Actually, the age range of participants in the study of Lesinski et al. (2016) is from 6 to 18 years. As such, they are rather youth athletes. The sentence was edited as follows:

Lesinski, Prieske and Granacher [13] conducted a meta-analysis on the effects of TRT on CoD performance in youth athletes aged between 6 and 18 years. ”

      94 Should that again be “…review and meta-analyze the literature ….”?

 Authors’ reply: Please, refer to our previous answer. Thank you

References         Be sure to check the correct journal notation of journal titles.  Some are capitalized and some are not. 

Authors’ reply: Thank you for your comment. We double-checked journals’ names for consistency.

Round 2

Reviewer 1 Report

The manuscript has been much improved and is acceptable.